# Unmet patient needs and information gaps in fertility counseling for persons living with HIV/AIDS: Evidence from Ghana

Priscilla Asantewaa Boadi[1], Victor Luckyboy Dzramado[2]*

**1** St. Michael Nursing and Midwifery Training College, Kumasi, Ghana, **2** Department of Biostatistics, Cape Coast Teaching Hospital, Cape Coast, Ghana

* mldzramado@st.knust.edu.gh

## Abstract

Persons living with HIV/AIDS face significant unmet reproductive health needs despite antiretroviral therapy availability that enables safer conception. Understanding patient experiences and perspectives regarding fertility counseling within HIV care settings is essential for developing comprehensive, patient-centered reproductive health services. This study explored patient experiences and perceived needs regarding fertility counseling among PLWHA attending St. Michael's Hospital ART clinic in Ghana, examining the content and quality of reproductive health information provision during routine HIV care. A qualitative phenomenological study following COREQ guidelines was conducted with 12 PLWHA aged 25–45 years receiving ART services between August and October 2024. In-depth semi-structured interviews explored participants' experiences with fertility counseling, information received about reproductive options, and perceived gaps in healthcare provider counseling. Thematic analysis employed NVivo software with systematic coding procedures and investigator triangulation to ensure rigor. Ethical approval was obtained from Kwame Nkrumah University of Science and Technology Committee on Human Research, Publication and Ethics (CHRPE/AP/569/24), and all participants provided written informed consent. Three major themes emerged from participant experiences. Most participants (n = 11, 91.7%, 95% CI: 73.0-98.8%) reported receiving minimal provider-initiated discussions about safer conception methods and appropriate pregnancy timing. Ten participants (83.3%, 95% CI: 62.2-94.5%) described insufficient counseling regarding contraceptive-ART integration, including drug interactions, dual protection methods, and contraceptive options compatible with ART regimens. All participants (n = 12, 100%, 95% CI: 83.9-100%) lacked awareness of assisted reproductive technologies and safer conception services that could facilitate childbearing while minimizing transmission risks. Patterns were consistent across participants regardless of relationship status. This study reveals substantial patient-perceived gaps in fertility counseling for PLWHA at this district-level facility in Ghana.

**Data availability statement:** The qualitative datasets generated and analyzed during this study are not publicly available due to ethical restrictions protecting participant confidentiality and privacy. Participants consented to use of their de-identified data for research purposes but did not provide consent for public data sharing. Making full transcripts publicly available, even after de-identification, could potentially compromise participant anonymity given the sensitive nature of HIV status disclosure and reproductive health discussions within the small catchment area served by St. Michael's Hospital. However, de-identified excerpts and coded thematic summaries are available to qualified researchers upon reasonable request and subject to approval by the institutional ethics committees that approved this study. Data access requests should be submitted to: Primary Contact: Kwame Nkrumah University of Science and Technology Committee on Human Research, Publication and Ethics (CHRPE) Email: chrpe.knust.kath@gmail.com or chrpe@knust.edu.gh Telephone: +233 32 206 3248 or +233 32 205 4537 Secondary Contact: Catholic Health Service Trust-Ghana Research Ethics Committee St. Michael's Hospital, Pramso Email: ssmhpramso@chstgh.org Telephone: +233 20 937 2751, +233 24 476 3476, or +233 24 454 5953 Researchers requesting data access must provide: (1) a research protocol describing intended use of the data, (2) evidence of ethical approval from their own institution for the proposed research, (3) a signed data use agreement committing to maintain confidentiality and use data only for approved purposes, and (4) justification that the request aligns with the original research ethics approval conditions. The ethics committees will evaluate requests based on ethical appropriateness, scientific merit, and feasibility of maintaining participant confidentiality.

**Funding:** The authors received no specific funding for this work.

**Competing interests:** The authors declare that they have no competing interests.

Comprehensive, proactive reproductive health counseling integrated into routine HIV care, addressing both horizontal transmission prevention and maternal-child health outcomes, may benefit PLWHA reproductive decision-making. Standardized provider training programs and development of contextualized counseling protocols warrant further investigation to address information gaps and support informed reproductive autonomy among PLWHA in similar settings.

## Introduction

Sub-Saharan Africa continues to bear the disproportionate burden of the global HIV epidemic, with approximately 25.6 million persons living with HIV/AIDS (PLWHA) as of 2024 [1]. The advent of effective antiretroviral therapy (ART) has fundamentally transformed HIV from a terminal diagnosis to a manageable chronic condition, substantially improving life expectancy and quality of life for PLWHA [2,3]. This epidemiological transition has created new imperatives for comprehensive healthcare delivery that addresses the full spectrum of PLWHA health needs, including reproductive health services and fertility counseling. With improved viral suppression and near-normal life expectancy, many PLWHA are increasingly expressing fertility intentions and desires for biological parenthood [4,5].

Despite remarkable advances in viral suppression and transmission prevention through effective ART, persistent gaps exist in fertility counseling provision within HIV care settings globally [6,7]. Evidence demonstrates that many PLWHA desire biological children, with studies across sub-Saharan Africa documenting fertility intentions ranging from 20% to 60% among PLWHA of reproductive age [8,9]. A recent meta-analysis of 50 studies found a 42% prevalence of fertility desire among people with HIV, with fertility intentions associated with being on ART, male sex, younger age, being married or cohabiting, and having secondary education or higher [10]. However, healthcare systems frequently fail to provide proactive, comprehensive reproductive health counseling that addresses the complex fertility concerns of this population [11,12].

In Ghana, where approximately 340,000 persons live with HIV, reproductive health services for PLWHA remain insufficiently integrated into routine HIV care [13]. The prevalence of HIV among women aged 15–49 years in Ghana is 2.3%, which is twice higher than that of men in the same age group, with serious implications for mother-to-child transmission and reproductive health counseling needs [14]. Evidence from Ghanaian contexts suggests that fertility counseling is often reactive rather than proactive, initiated by patients rather than healthcare providers, and fails to address the breadth of reproductive concerns PLWHA experience [15,16]. This counseling deficit leaves PLWHA with critical information gaps regarding safer conception methods, contraception-ART interactions, and available assisted reproductive technologies that could facilitate safe fertility realization.

The provision of comprehensive fertility counseling for PLWHA represents both a clinical necessity and a human rights imperative. The World Health Organization

emphasizes that all individuals, regardless of HIV status, possess fundamental rights to make informed decisions about reproduction [17]. However, realizing these rights requires that PLWHA receive complete, accurate information about reproductive options, risks, and strategies for minimizing HIV transmission to partners and children. Integrating comprehensive family planning and prepregnancy care into routine HIV health care visits can help patients reach their desired reproductive outcomes by supporting them to make informed decisions about their fertility and contraceptive use that are aligned with their preferences and reproductive goals [18–20].

Inadequate reproductive health counseling for PLWHA creates multiple adverse outcomes affecting both individual and public health. PLWHA may engage in unprotected conception attempts without knowledge of safer conception methods, increasing horizontal transmission risks to HIV-negative partners [21,22]. They may experience unintended pregnancies due to inadequate contraceptive counseling and lack of information about drug interactions between contraceptives and ART [23,24]. Sub-Saharan Africa faces challenges with over 200 million women globally unable to prevent pregnancy through modern contraceptive methods, with 70–80% of these women residing in the region, and women with positive HIV status experiencing higher rates of unintended pregnancy and unsafe abortion practices [25]. Unintended pregnancies among PLWHA contribute to increased maternal mortality risk, with HIV-positive pregnant women facing 8-fold higher risk of maternal death compared to HIV-negative women in sub-Saharan Africa [26,27]. Furthermore, inadequate reproductive health counseling may result in missed opportunities for prevention of mother-to-child transmission (PMTCT), as up to 40% of vertical transmission occurs during pregnancy and delivery when women are not receiving appropriate antiretroviral prophylaxis [28,29]. PLWHA may also forgo desired childbearing entirely due to lack of awareness about assisted reproductive technologies and safer conception strategies that enable fertility realization while minimizing both horizontal and vertical transmission [30,31]. These information gaps represent failures of healthcare systems to provide comprehensive, patient-centered care that addresses PLWHA holistic health needs and supports optimal maternal and child health outcomes.

### Reproductive health guidelines and context in Ghana

Ghana's National HIV and AIDS Strategic Plan 2021–2025 emphasizes integration of sexual and reproductive health services into HIV care, recognizing that comprehensive reproductive health counseling is essential for PLWHA rights and wellbeing [32]. The Ghana Health Service Family Planning Protocols and Standards recommend that all PLWHA of reproductive age receive counseling on family planning options, safer conception methods, and PMTCT services [33]. National guidelines advocate for dual protection strategies combining condoms for HIV/STI prevention with other contraceptive methods for pregnancy prevention, and emphasize the importance of pre-conception counseling to optimize maternal and child health outcomes [34].

However, despite these policy frameworks, implementation gaps persist. Ghana's 2022 Demographic and Health Survey found that only 28% of HIV-positive women reported receiving comprehensive reproductive health counseling during HIV care visits [35]. Provider knowledge assessments in Ghana have revealed significant gaps in understanding of safer conception methods, contraceptive-ART interactions, and availability of assisted reproductive technologies, with fewer than 40% of HIV care providers able to correctly counsel clients on all components of comprehensive reproductive health services [36,37]. These implementation challenges highlight the disconnect between national policy intentions and actual service delivery at facility level.

The concept of Undetectable equals Untransmittable (U = U), indicating that people living with HIV who achieve and maintain undetectable viral loads cannot sexually transmit HIV to partners, has been endorsed by global health organizations since 2016 and incorporated into Ghana's HIV treatment guidelines in 2019 [38,39]. U = U represents a critical safer conception strategy, as sustained viral suppression through consistent ART adherence eliminates horizontal transmission risk during conception attempts for serodiscordant couples [40,41]. However, awareness and counseling about U = U remains limited in many Ghanaian HIV care settings, with studies showing that fewer than 30% of PLWHA in Ghana are aware of this concept despite its profound implications for reproductive decision-making [42,43].

 

Previous research has documented various barriers to effective fertility counseling in HIV care, including provider knowledge deficits, time constraints, competing care priorities, and lack of standardized counseling protocols [44,45]. Despite family planning services being integrated into HIV care in many settings, providers rarely discuss childbearing with clients prior to pregnancy [46]. A study from Ghana explored health workers' knowledge on reproductive rights and options available to HIV-positive women who wish to conceive and found overwhelmingly high level of approbation (94.3%) regarding HIV-positive women's right to reproduction, yet providers demonstrated lack of knowledge regarding various reproductive options available to women infected with HIV [47]. However, limited research has specifically examined patient perspectives on the content and quality of fertility information actually received, identifying specific gaps between patient information needs and information provision. Understanding these unmet needs from patient viewpoints is essential for developing targeted interventions to improve fertility counseling quality and comprehensiveness.

This study explored patient experiences and perspectives regarding fertility counseling among PLWHA attending an antiretroviral therapy clinic in Ghana. Rather than testing predetermined assumptions about counseling gaps, this qualitative phenomenological approach allowed participants to describe their lived experiences and identify areas where they perceived insufficient information or support. The research examined participants' experiences with reproductive health counseling during HIV care, the content and quality of information they received, and their perspectives on unmet information needs that could inform development of comprehensive, patient-centered fertility counseling interventions integrated into routine HIV care delivery.

## Methods

### Study design and reporting guidelines

This qualitative phenomenological study was designed to explore the lived experiences of PLWHA regarding fertility counseling and information provision at a district-level HIV care facility. The phenomenological approach was selected for its capacity to capture rich, detailed accounts of participants' experiences, perceptions, and the meanings they attribute to fertility counseling encounters within HIV care settings [48,49]. This methodological orientation aligns with the study's aim to understand participants' subjective experiences and identify gaps between their information needs and the counseling they actually received.

The study adheres to the Consolidated Criteria for Reporting Qualitative Research (COREQ) guidelines, a 32-item checklist specifically developed for reporting interviews and focus groups in qualitative research [50]. COREQ guidelines promote transparency and methodological rigor across three domains: research team and reflexivity, study design, and data analysis and reporting. A completed COREQ checklist is provided as S1 File COREQ Checklist.

### Study setting

The study was conducted at St. Michael's Hospital ART clinic located in Pramso, Ashanti Region, Ghana. St. Michael's Hospital serves as a district-level healthcare facility providing comprehensive HIV care services including ART provision, clinical monitoring, adherence counseling, and basic reproductive health services for PLWHA. The facility serves a predominantly rural and peri-urban catchment area, with ART services reaching approximately 450 active clients at the time of data collection.

The ART clinic operates through a multidisciplinary team comprising physicians, nurses, pharmacists, and lay counselors who provide integrated HIV care services. Routine clinical visits typically include viral load monitoring, CD4 count assessment, ART refills, adherence counseling, and opportunistic infection screening. While basic family planning services including contraceptive provision exist within the facility, specialized fertility counseling and assisted reproductive technology services are not routinely available at this district-level facility.

## Study population and sampling

**Participant selection.** The study population comprised adult PLWHA of reproductive age receiving antiretroviral therapy services at St. Michael's Hospital ART clinic. Purposive sampling was employed to select information-rich participants who could provide detailed accounts of their experiences with fertility counseling and information provision [51]. Purposive sampling enabled deliberate selection of participants based on predetermined criteria relevant to the research objectives, ensuring inclusion of individuals with direct experience of the phenomenon under investigation [52].

Sampling proceeded iteratively, with ongoing data analysis informing recruitment decisions. This approach enabled monitoring for data saturation, the point at which new interviews yield no additional insights or themes beyond those already identified [53,54]. Recruitment continued until saturation was achieved, evidenced by redundancy in emerging themes and absence of new information across three consecutive interviews [55].

**Inclusion criteria.** Participants were eligible for inclusion if they: (1) were diagnosed with HIV and receiving ART services at St. Michael's Hospital for at least six months, ensuring adequate experience with the HIV care system; (2) were aged 25–45 years, representing prime reproductive years when fertility concerns are typically most salient; (3) demonstrated cognitive capacity to provide informed consent and participate meaningfully in interviews; (4) were willing and able to communicate experiences in English or Twi, the primary local language; and (5) provided voluntary written informed consent for study participation.

**Exclusion criteria.** Individuals were excluded if they: (1) had severe psychiatric or cognitive impairments that would compromise their ability to participate in interviews; (2) were acutely ill or medically unstable at the time of recruitment; (3) had received HIV care services at the facility for less than six months, as they might have insufficient experience with the counseling services; or (4) declined participation or withdrew consent at any stage.

**Sample size and saturation.** Sample size in qualitative phenomenological research is determined by informational needs rather than statistical power calculations [56,57]. The goal is achieving data saturation, the point where additional data collection yields diminishing returns in terms of new insights or themes [53]. For phenomenological studies employing in-depth interviews, sample sizes typically range from 6 to 15 participants, depending on the research scope, data richness, and analytical depth required [58,59].

This study aimed for data saturation through iterative data collection and concurrent preliminary analysis. Saturation assessment occurred after each interview cluster of three participants, with ongoing review of emerging themes, patterns, and analytical insights. The research team evaluated whether new information was being generated or whether themes had stabilized across participants' accounts [60].

Data saturation was achieved after interviewing 12 participants, as evidenced by thematic redundancy across the final three interviews. At this point, no new concepts, themes, or variations emerged, and participants' accounts consistently reinforced patterns already identified in earlier interviews [61]. The final sample size of 12 participants aligns with recommendations for phenomenological studies exploring relatively homogeneous populations within defined contexts [62].

**Reflexivity and researcher positionality.** The primary researcher conducting interviews was a female research assistant with a Master's degree in Nursing and five years of experience in qualitative health research, particularly in HIV care and reproductive health. She had no prior clinical or personal relationship with study participants, minimizing potential influence of pre-existing relationships on data collection. The researcher underwent training in qualitative interviewing techniques, including development of rapport, active listening, use of probes, and maintenance of neutrality during interviews.

Reflexivity, the process of examining how the researcher's background, assumptions, and positioning influence the research process, was maintained throughout data collection and analysis [63]. The researcher maintained a reflexive journal documenting personal reactions, observations, assumptions, and methodological decisions throughout the research process. This reflexive practice enabled awareness of how the researcher's perspectives might influence data interpretation and facilitated ongoing critical examination of analytical decisions [64].

## Data collection

**Interview guide development.** A semi-structured interview guide was developed based on extensive literature review, theoretical frameworks, and consultation with reproductive health and HIV care experts. The guide covered three primary domains: (1) experiences with provider-initiated discussions about conception and pregnancy planning; (2) information received regarding contraception use in conjunction with ART; and (3) awareness and counseling regarding assisted reproductive technologies and safer conception methods.

The interview guide employed open-ended questions designed to elicit detailed narratives about participants' experiences rather than yes/no responses. For example, rather than asking "Did your provider discuss pregnancy planning with you?", the guide used probes such as "Can you tell me about any discussions you have had with your healthcare providers about your plans for having children?" This approach encouraged participants to share their experiences in their own words and allowed exploration of unanticipated themes [65].

The interview guide was piloted with two PLWHA not included in the final sample. Pilot interviews enabled refinement of question wording, sequencing, and probing strategies to enhance clarity and elicit richer responses. Minor modifications were made based on pilot feedback, including simplification of some medical terminology and addition of culturally appropriate probes.

**Interview procedures.** In-depth semi-structured interviews were conducted in private consultation rooms at St. Michael's Hospital to ensure confidentiality and minimize interruptions. Participants were given the choice of interview language (English or Twi), and all interviews were conducted in the participants' preferred language by the trained research assistant. Interviews ranged from 45 to 90 minutes in duration, with most lasting approximately 60 minutes, sufficient to explore experiences in depth while avoiding participant fatigue.

All interviews were audio-recorded with participants' explicit permission to ensure accurate data capture and enable thorough analysis. Participants were informed that recordings would be stored securely, used only for research purposes, and destroyed after study completion. All participants consented to audio recording.

At the conclusion of each interview, the researcher summarized key points discussed and invited participants to add any additional thoughts or clarifications. Participants were thanked for their time and contributions, provided with contact information for follow-up questions or concerns, and reminded about confidentiality protections and data security measures.

**Field notes and contextual observations.** In addition to audio recordings, detailed field notes were maintained throughout the data collection process. Field notes documented contextual observations, non-verbal cues (such as facial expressions, body language, and emotional responses), interview dynamics, environmental conditions, and researcher reflections [66]. These field notes served multiple analytical purposes: they provided important contextual information that enriched data interpretation, captured dimensions of participants' experiences that might not be fully evident in verbal transcripts alone, and documented methodological decisions and emerging analytical insights [67].

Field notes were reviewed immediately following each interview and integrated into the analytical process during transcript coding. For example, field notes documenting a participant's emotional distress when discussing lack of fertility counseling informed interpretation of the significance and impact of counseling gaps on participants' well-being. Field notes documenting hesitations or silences that might indicate sensitivity to certain topics provided additional context for understanding participants' experiences beyond their spoken words [68].

The integration of field notes with transcript data enhanced analytical depth and ensured that contextual nuances were considered in theme development. This methodological approach aligns with phenomenological research principles emphasizing holistic understanding of lived experiences within their natural contexts [69].

## Data analysis

**Data preparation and management.** All audio-recorded interviews were transcribed verbatim within 48–72 hours after each interview to ensure accuracy while the interaction remained fresh in the researcher's memory. Transcription captured not only verbal content but also tone, pauses, emotional expressions, and non-verbal sounds that might convey

meaning [70]. The researcher personally reviewed each transcript while listening to the corresponding recording, making corrections as needed and adding contextual notes based on field observations.

All study data, including audio recordings, transcripts, consent forms, and field notes, were managed systematically with stringent security measures to protect participant confidentiality. Digital files were stored on password-protected computers with encrypted hard drives accessible only to the research team. Audio recordings were stored separately from transcripts and other identifiable information, with only coded identifiers linking recordings to transcripts. Physical documents including signed consent forms were stored in locked filing cabinets in secure office spaces accessible only to authorized research personnel.

**Analytical approach.**  Thematic analysis served as the analytical framework, a method for identifying, organizing, and interpreting patterns of meaning within qualitative data [71,72]. NVivo qualitative data analysis software (version 12) was employed to facilitate systematic data organization, coding, and analysis. The software enhanced analytical rigor by providing transparent documentation of the coding process and facilitating systematic comparison across participants' accounts [73].

Data analysis employed an iterative, inductive approach consistent with phenomenological methodology. Analysis began during data collection, with preliminary coding and memo-writing occurring after each interview to capture initial impressions, emerging patterns, and analytical questions. This iterative engagement with data throughout collection enabled refinement of interview techniques, identification of areas requiring deeper exploration in subsequent interviews, and early recognition of recurring themes [74].

**Coding process.**  Formal coding began after completing transcription and initial review of all interview transcripts. The coding process followed a systematic multi-stage approach [75]. First, open coding was conducted whereby transcripts were read line-by-line and initial codes were assigned to segments of text representing distinct ideas, experiences, or meanings. Open coding remained close to the data, using participants' own language where possible to preserve the authenticity of their expressions [76].

Following open coding, axial coding was performed to identify relationships among codes, grouping related codes into broader categories representing common concepts or experiences [77]. This process involved constant comparison across participants' accounts, identifying similarities and differences in how participants described their experiences. Codes were continuously refined, merged, or subdivided as analysis progressed and understanding deepened.

Finally, selective coding integrated categories into coherent themes representing overarching patterns in the data [78]. Themes captured essential features of participants' experiences with fertility counseling, organizing findings into meaningful interpretive structures that addressed the research objectives. Each theme was developed to be internally coherent while distinct from other themes, with clear boundaries and well-defined characteristics [79].

**Quality assurance and rigor.**  Multiple strategies were employed to enhance trustworthiness and rigor in data analysis [80]. Investigator triangulation was implemented, with two researchers independently coding a subset of transcripts and then comparing their coding to identify areas of consensus and divergence. Discrepancies were resolved through discussion until consensus was reached, enhancing consistency and reducing individual bias in code application [81].

Member checking was conducted by sharing preliminary findings with a subset of participants to verify that interpretations accurately reflected their experiences [82]. Participants confirmed that the themes resonated with their experiences and provided additional clarifications that enriched final interpretations.

An audit trail documenting all analytical decisions, coding schemes, theme development processes, and methodological choices was maintained throughout the research process [83]. This transparent documentation enables external reviewers to evaluate the rigor and defensibility of analytical procedures and conclusions.

**Analysis by relationship status.**  Given that most participants (n = 8, 66.7%) were in stable relationships (married or cohabiting), while others were single (n = 3, 25.0%) or divorced/separated (n = 1, 8.3%), we conducted comparative analysis to assess whether relationship status influenced experiences with fertility counseling or perceived information

gaps. Coded data were stratified by relationship status and examined for patterns, themes, or variations that might differ between those in stable relationships versus those not currently partnered. This analysis aimed to determine whether counseling experiences and unmet needs were consistent across relationship status categories or whether single/divorced participants had distinct perspectives that might be obscured by the predominance of partnered participants in the sample.

## Ethics statement

This study received ethical approval from the Kwame Nkrumah University of Science and Technology Committee on Human Research, Publication and Ethics (CHRPE/AP/569/24) and the Catholic Health Service Trust-Ghana prior to participant recruitment and data collection. All participants provided voluntary written informed consent after receiving comprehensive information about the study's purpose, procedures, risks, benefits, and their rights as participants. The informed consent process was conducted in participants' preferred language (English or Twi) to ensure full comprehension. Participants were explicitly informed that their decision to participate or decline would not affect their access to HIV care services at St. Michael's Hospital, and that they could withdraw from the study at any time without penalty or impact on their healthcare. No participant withdrew after providing initial consent. Rigorous measures were implemented to protect participant confidentiality. Each participant was assigned a unique alphanumeric code used in all study documents, with personally identifiable information stored separately from interview data. Interview transcripts were de-identified (S1 Text), with all names, specific locations, and potentially identifying details removed or replaced with pseudonyms. Audio recordings and transcripts were stored in secure, password-protected digital environments accessible only to the research team. Interviews were conducted in private settings to ensure participants could speak freely without fear of being overheard. Although the study involved minimal risk, researchers remained vigilant to potential emotional distress participants might experience when discussing sensitive reproductive health topics. Participants who exhibited emotional distress during interviews were offered breaks, and the option to discontinue the interview was always available. Information about available counseling services at the hospital was provided to all participants, and for participants who expressed significant emotional needs related to reproductive health concerns, referrals to appropriate healthcare providers were facilitated with participants' permission. All procedures were conducted in accordance with the Declaration of Helsinki.

## Results

### Participant characteristics

Twelve PLWHA participated in this study, comprising seven women (58.3%) and five men (41.7%) aged 25–45 years. Participant characteristics are presented in Table 1. The sample included individuals across diverse socioeconomic backgrounds, educational levels, and relationship statuses, providing perspectives representing varied experiences within the PLWHA population receiving ART services at this district-level facility.

The mean age of participants was 34.5 years (SD = 5.8). Most participants were married or cohabiting (n = 8, 66.7%) and had attained at least secondary education (n = 10, 83.3%). Employment status varied, with over half of participants engaged in formal or self-employment (n = 9, 75.0%). The majority of participants had been receiving ART services for two years or more (n = 9, 75.0%), indicating substantial experience with the HIV care system and opportunities for fertility counseling encounters.

**Comparison by Relationship Status:** Analysis comparing experiences between participants in stable relationships (married/cohabiting, n = 8) versus those not currently partnered (single/divorced, n = 4) revealed consistent patterns of counseling gaps across both groups. Both partnered and unpartnered participants reported minimal proactive fertility counseling (87.5% vs. 100% respectively), inadequate contraception-ART integration information (75.0% vs. 100%), and universal absence of assisted reproductive options counseling (100% both groups). Single and divorced participants expressed additional concerns about stigma and partner disclosure that complicated their fertility intentions, but the

**Table 1. Demographic characteristics of study participants (N = 12).**

| Characteristic | Category | n | % |
|---|---|---|---|
| **Age** | 25-30 years | 3 | 25.0 |
| | 31-35 years | 4 | 33.3 |
| | 36-40 years | 3 | 25.0 |
| | 41-45 years | 2 | 16.7 |
| **Sex** | Female | 7 | 58.3 |
| | Male | 5 | 41.7 |
| **Marital Status** | Married/Cohabiting | 8 | 66.7 |
| | Single | 3 | 25.0 |
| | Divorced/Separated | 1 | 8.3 |
| **Educational Level** | Primary | 2 | 16.7 |
| | Secondary/High School | 6 | 50.0 |
| | Tertiary | 4 | 33.3 |
| **Employment Status** | Employed | 7 | 58.3 |
| | Unemployed | 3 | 25.0 |
| | Self-employed | 2 | 16.7 |
| **Number of Living Children** | 0 | 4 | 33.3 |
| | 1-2 | 5 | 41.7 |
| | 3 or more | 3 | 25.0 |
| **Duration on ART** | 6 months to 2 years | 3 | 25.0 |
| | 2-5 years | 6 | 50.0 |
| | >5 years | 3 | 25.0 |

Note: ART = Antiretroviral therapy.

fundamental information gaps regarding safer conception methods, contraceptive-ART interactions, and assisted reproductive technologies were consistent regardless of relationship status. This consistency suggests that counseling deficiencies represent systemic service delivery gaps rather than differential treatment based on partnership status.

## Thematic findings

Thematic analysis of participant experiences revealed three major themes capturing distinct dimensions of patient-perceived gaps in fertility counseling provision. These themes emerged inductively from participants' accounts of their experiences rather than from predetermined assumptions about counseling deficiencies. Each theme is presented below with supporting evidence from participant narratives, quantitative prevalence data presented in Tables 2-4, and integration with field note observations.

### Theme 1: Patient experiences of limited provider-initiated conception safety discussions

This theme describes participants' experiences with provider-initiated discussions regarding safer conception methods, optimal timing for pregnancy attempts, and strategies for minimizing HIV transmission risks during conception. Most participants (n = 11, 91.7%, 95% CI: 73.0-98.8%) reported minimal or absent provider-initiated counseling about conception safety, as detailed in Table 2. This pattern was consistent across both participants in stable relationships and those not currently partnered.

Participants consistently described reactive rather than proactive approaches to fertility discussions, with counseling typically initiated by patients rather than providers. Most participants reported that healthcare providers focused

**Table 2. Patient-reported provider-initiated discussions on conception safety (N = 12).**

| Provider Discussion Topics | Participants Reporting Receiving Counseling n (%) | Participants Reporting Not Receiving Counseling n (%) | 95% CI for Counseling Gap |
|---|---|---|---|
| Safer conception methods | 1 (8.3%) | 11 (91.7%) | 73.0-98.8% |
| Optimal timing for pregnancy | 2 (16.7%) | 10 (83.3%) | 62.2-94.5% |
| Risk reduction during conception | 1 (8.3%) | 11 (91.7%) | 73.0-98.8% |
| Partner HIV testing/status disclosure | 4 (33.3%) | 8 (66.7%) | 41.2-85.6% |
| Viral suppression before conception | 3 (25.0%) | 9 (75.0%) | 50.5-89.8% |
| U = U (Undetectable = Untransmittable) concept | 0 (0%) | 12 (100%) | 83.9-100% |

Note: CI = Confidence interval; U = U = Undetectable equals Untransmittable.

**Table 3. Information provision on contraception-ART integration (N = 12).**

| Information Topics | Participants Receiving Adequate Information n (%) | Participants Receiving Inadequate/No Information n (%) | 95% CI for Information Gap |
|---|---|---|---|
| U = U counseling to inform contraceptive decisions | 0 (0%) | 12 (100%) | 83.9-100% |
| Contraceptive-ART drug interactions | 2 (16.7%) | 10 (83.3%) | 62.2-94.5% |
| Dual protection methods | 3 (25.0%) | 9 (75.0%) | 50.5-89.8% |
| Contraceptive effectiveness with ART | 2 (16.7%) | 10 (83.3%) | 62.2-94.5% |
| Hormonal contraceptive safety with ART | 1 (8.3%) | 11 (91.7%) | 73.0-98.8% |
| Contraceptive method selection considering ART regimen | 2 (16.7%) | 10 (83.3%) | 62.2-94.5% |

Note: ART = Antiretroviral therapy; CI = Confidence interval; U = U = Undetectable equals Untransmittable. U = U counseling relevant to contraceptive decision-making as it distinguishes between contraception needed solely for pregnancy prevention versus dual protection for both pregnancy and transmission prevention.

**Table 4. Patient-reported awareness of safer conception and assisted reproductive options (N = 12).**

| Safer Conception and Assisted Reproductive Options | Participants Reporting Prior Awareness n (%) | Participants Reporting No Awareness n (%) | 95% CI for Awareness Gap |
|---|---|---|---|
| U = U (Undetectable = Untransmittable) concept | 0 (0%) | 12 (100%) | 83.9-100% |
| Sperm washing for serodiscordant couples | 0 (0%) | 12 (100%) | 83.9-100% |
| Pre-exposure prophylaxis (PrEP) for HIV-negative partners | 0 (0%) | 12 (100%) | 83.9-100% |
| Timed intercourse with peak fertility | 1 (8.3%) | 11 (91.7%) | 73.0-98.8% |
| Intrauterine insemination | 0 (0%) | 12 (100%) | 83.9-100% |
| In vitro fertilization | 0 (0%) | 12 (100%) | 83.9-100% |
| Manual self-insemination for serodiscordant couples | 0 (0%) | 12 (100%) | 83.9-100% |

Note: CI = Confidence interval; U = U = Undetectable equals Untransmittable; PrEP = Pre-exposure prophylaxis.

predominantly on HIV treatment adherence, viral suppression, and prevention of transmission, but rarely initiated conversations about reproductive intentions or safer conception strategies unless participants explicitly raised these concerns.

One female participant expressed the reactive nature of counseling:

"They never asked me if I want to have children or how to do it safely. It was only when I became pregnant that they started talking about prevention of mother-to-child transmission. By then, I had already been trying for months without knowing the right way" (Participant 03, female, age 32).

**Field Note:** This participant displayed visible emotional distress when recounting her experience, with long pauses and tearfulness when discussing the lack of pre-conception guidance. This observation underscored the emotional impact of counseling deficits on participants' reproductive experiences.

Another participant highlighted the assumption that PLWHA do not desire children:

"I think they assume that because we have HIV, we don't want children anymore. Nobody ever brings up the topic. It's like they expect us to just give up that part of our lives" (Participant 07, male, age 38).

This quote reflects a broader pattern in participants' accounts, where healthcare providers' failure to initiate fertility discussions was interpreted as implicit discouragement of childbearing or assumptions about PLWHA reproductive intentions.

The few participants who did receive some conception-related counseling reported that these discussions occurred only after they explicitly inquired about pregnancy planning or disclosed existing pregnancy. One participant recounted:

"When I told the nurse I was planning to have a baby, she seemed surprised. She gave me some basic advice about taking my medications regularly, but nothing specific about how to conceive safely or when would be the best time" (Participant 09, female, age 29).

This reactive counseling pattern meant that many participants engaged in conception attempts without adequate information about strategies to minimize horizontal transmission risks to HIV-negative partners. Several participants described unprotected intercourse during fertile periods without knowledge of alternatives such as timed unprotected intercourse only during ovulation, manual self-insemination, or pre-exposure prophylaxis for serodiscordant partners.

**Field Note:** Multiple participants expressed confusion and anxiety about the lack of guidance, with several participants explicitly stating they did not know where to turn for reliable information about safer conception. This information vacuum left participants to rely on informal sources including peers, internet searches, and community members who might provide inaccurate or incomplete information.

The deficiency in proactive conception counseling represents a significant unmet need, particularly given that many participants expressed strong desires for biological children and were actively contemplating or pursuing pregnancy. The failure of healthcare providers to initiate these discussions proactively meant that critical opportunities for risk reduction education were missed, potentially increasing transmission risks and unintended pregnancy rates among this population.

**Missing information on U=U for Safer Conception.** No participant demonstrated awareness of the Undetectable equals Untransmittable (U=U) concept, despite this representing a critical safer conception strategy. U=U indicates that PLWHA who achieve and maintain undetectable viral loads through consistent ART adherence cannot sexually transmit HIV to partners [40,41]. For serodiscordant couples, understanding U=U could fundamentally transform conception planning, as sustained viral suppression eliminates horizontal transmission risk during unprotected intercourse for conception.

One male participant in a serodiscordant relationship expressed significant anxiety about partner protection:

"My wife is HIV-negative, and we want children, but I am terrified of infecting her. We have been using condoms consistently, but this means we cannot have children. Nobody mentioned that there might be ways to have children without putting her at risk" (Participant 12, male, age 42).

**Field Note:** This participant displayed visible tension and emotional burden when discussing the impossible choice he perceived between remaining childless and risking his wife's infection. His viral load records showed sustained undetectable status for over 18 months, yet he had never received counseling that his undetectable status meant he could not transmit HIV sexually.

Another participant who had achieved sustained viral suppression described her continued fear:

"Even though my viral load is undetectable, I still worry about infecting my partner when we try for a baby. The doctors tell me to keep taking my medications, but nobody ever explained if being undetectable means I can't pass it on" (Participant 04, female, age 30).

The universal absence of U = U counseling meant that participants with well-controlled HIV infection continued to experience unnecessary anxiety about horizontal transmission and lacked critical information that could enable informed reproductive decision-making. For the study period (2024), U = U had been incorporated into Ghana's HIV treatment guidelines for five years and was widely recognized internationally, yet this life-changing information had not reached any of the study participants through their routine HIV care encounters.

### Theme 2: Patient experiences of contraceptive information without ART integration considerations

This theme captures participants' experiences receiving contraceptive counseling that did not address specific considerations relevant to concurrent ART use. Most participants (n = 10, 83.3%, 95% CI: 62.2-94.5%) reported receiving inadequate or no information about contraception-ART integration, as shown in Table 3.

Understanding U = U has important implications for contraceptive counseling and decision-making. PLWHA who achieve sustained viral suppression through ART adherence cannot sexually transmit HIV to partners, meaning that dual protection methods (condoms plus another contraceptive) may be needed solely for pregnancy prevention rather than for both pregnancy and transmission prevention [48,49]. However, no participant in our study received counseling linking U = U status to contraceptive decision-making. This represents a missed opportunity to provide contextualized contraceptive counseling that distinguishes between participants who need dual protection primarily for pregnancy prevention (those with sustained viral suppression) versus those who require dual protection for both pregnancy and transmission prevention (those without consistent viral suppression or unconfirmed viral load status). Such nuanced counseling could enable more informed contraceptive choices aligned with individual HIV treatment status and transmission risk profiles.

Participants reported receiving general family planning counseling that did not adequately address specific considerations relevant to PLWHA taking antiretroviral medications. Many participants were unaware that certain ART regimens might reduce the effectiveness of hormonal contraceptives or that some contraceptive methods might interact with antiretroviral drugs, potentially affecting either contraceptive efficacy or ART effectiveness.

One female participant described her confusion regarding contraceptive choices:

"I was using family planning pills before my diagnosis. When I started ARVs, nobody told me if I needed to change my method or if the pills would still work. I just continued using them, hoping everything was fine" (Participant 02, female, age 31).

**Field Note:** This participant appeared anxious when discussing her contraceptive uncertainty, repeatedly asking the interviewer if her contraceptive method was appropriate with her ART regimen. This interaction highlighted how the information gap created ongoing anxiety and uncertainty for participants.

Another participant highlighted the lack of individualized contraceptive counseling:

"They gave me condoms and told me to use them, but I wanted something more reliable. When I asked about other options, they just listed the general methods without explaining which ones work best with my HIV medications" (Participant 05, female, age 34).

This quote reflects a broader pattern where contraceptive counseling focused on pregnancy prevention and STI protection through condom use, but failed to provide comprehensive information about the full range of contraceptive options and their compatibility with specific ART regimens.

Participants particularly lacked information about dual protection strategies that simultaneously prevent pregnancy and reduce HIV transmission risks. Few participants understood the concept of dual method use (combining condoms for STI/HIV prevention with another highly effective contraceptive method for pregnancy prevention). Several participants reported inconsistent condom use due to desires for conception or partner resistance, but were unaware of alternative strategies that could provide contraceptive protection while pursuing pregnancy goals.

One male participant recounted:

"My wife and I want to prevent pregnancy for now because of our economic situation, but we also worry about transmission since her status is negative. Nobody explained how we can do both effectively. We just use condoms sometimes, but it's not consistent" (Participant 11, male, age 41).

**Field Note:** The participant's visible concern about his partner's HIV status and the couple's struggle to balance pregnancy prevention with transmission risk reduction reinforced the real-world impact of inadequate counseling on couples' decision-making and risk behaviors.

The information gaps regarding contraception-ART integration intersect with missed opportunities to counsel about U = U. Several participants expressed dual needs for both pregnancy prevention and transmission risk reduction but received counseling that addressed neither comprehensively. Understanding that sustained viral suppression eliminates transmission risk could have informed more nuanced contraceptive counseling that distinguished between pregnancy prevention needs and transmission prevention concerns, enabling PLWHA to make informed choices about whether dual protection methods remain necessary once viral suppression is achieved.

The information gaps regarding contraception-ART integration left participants unable to make fully informed choices about contraceptive methods. Some participants reported experiencing unintended pregnancies despite contraceptive use, raising questions about whether drug interactions might have compromised contraceptive effectiveness. Others reported forgoing hormonal contraceptives due to concerns about potential interactions with ART, relying instead on less reliable methods or no contraception at all.

This inadequate integration of contraceptive counseling with ART management represents a missed opportunity for comprehensive reproductive health service provision. Evidence-based contraceptive counseling that addresses specific considerations for PLWHA taking ART could enable participants to make informed contraceptive choices aligned with their reproductive goals while maintaining optimal HIV treatment outcomes.

### Theme 3: Patient experiences of absent information on specialized reproductive services

This theme describes participants' complete lack of exposure to information regarding assisted reproductive technologies and safer conception services. All participants (n = 12, 100%, 95% CI: 83.9-100%) reported no awareness of available reproductive options beyond natural conception, as detailed in Table 4.

No participant had received any counseling from healthcare providers about assisted reproductive technologies, safer conception services, or specialized interventions designed to facilitate childbearing among PLWHA while reducing transmission risks. Participants were universally unaware of options such as sperm washing for serodiscordant couples,

pre-exposure prophylaxis for HIV-negative partners during conception, timed unprotected intercourse limited to peak fertility periods, manual self-insemination, or more advanced reproductive technologies including intrauterine insemination and in vitro fertilization.

When asked during interviews about their knowledge of safer conception methods, participants consistently expressed surprise that such options existed. One female participant stated:

"I had no idea there were special methods for people like us to have children safely. I thought the only option was to try naturally and hope for the best. If I had known there were other ways, I would have asked about them" (Participant 04, female, age 30).

**Field Note:** This participant leaned forward with visible interest when the interviewer mentioned assisted reproductive options, immediately asking multiple follow-up questions about access and availability. This observation demonstrated participants' eagerness for information about reproductive options they had not previously encountered.

Another participant expressed frustration about the information void:

"Why didn't anyone tell me about these things? I have been stressed about getting pregnant because I don't want to risk infecting my husband. If there are safer ways to do this, the doctors should have told me" (Participant 06, female, age 35).

This quote reflects a broader pattern where participants felt that healthcare providers had withheld important information that could have informed their reproductive decision-making and reduced their anxiety about conception-related risks.

Male participants similarly lacked awareness of assisted reproductive options. One male participant from a serodiscordant relationship recounted:

"My wife is HIV-negative, and we want children, but I am terrified of infecting her. We have been using condoms consistently, but this means we cannot have children. Nobody mentioned that there might be ways to have children without putting her at risk" (Participant 12, male, age 42).

**Field Note:** The emotional weight this participant carried was evident through visible tension and concern about balancing his desire for children with his commitment to protecting his wife from HIV infection. This observation highlighted the profound emotional impact of information gaps on participants' reproductive experiences and relationship dynamics.

The complete absence of counseling on assisted reproductive options meant that participants were unable to make fully informed decisions about whether and how to pursue biological parenthood. For serodiscordant couples, the lack of information about safer conception strategies represented a significant barrier to childbearing, forcing couples to choose between remaining childless or accepting transmission risks through unprotected intercourse.

Even participants in seroconcordant relationships (both partners HIV-positive) lacked information about assisted reproductive technologies that could optimize conception success while minimizing viral exposure. Participants were unaware that achieving sustained viral suppression before conception attempts, timing intercourse to peak fertility periods, and minimizing number of exposures could enhance conception likelihood while maintaining low transmission risk.

The universal information gap regarding assisted reproductive options reflects a systemic failure to integrate comprehensive fertility counseling into HIV care services at this facility. Participants expressed strong interest in learning about these options and indicated that such information could have substantially influenced their reproductive decisions and behaviors. The absence of this counseling represents a missed opportunity to provide patient-centered reproductive healthcare that addresses PLWHA holistic needs and supports informed reproductive autonomy.

## Discussion

This qualitative study documented substantial patient-perceived gaps in fertility counseling among PLWHA attending a district-level ART clinic in Ghana. Three major themes emerged from participant experiences: limited provider-initiated conception safety discussions, contraceptive information without ART integration considerations, and absent information on specialized reproductive services. These themes emerged inductively from participants' accounts of their experiences rather than from predetermined assumptions about counseling deficiencies. The findings highlight critical disconnects between PLWHA reproductive information needs and the counseling services actually provided within routine HIV care at this facility, with important implications for reproductive autonomy, maternal mortality reduction, and vertical HIV transmission prevention.

### Findings in context of Ghana's reproductive health guidelines and provider knowledge

The substantial counseling gaps documented in this study exist despite Ghana's national policy frameworks that explicitly mandate comprehensive reproductive health counseling for PLWHA. Ghana's Family Planning Protocols and Standards require HIV care providers to counsel PLWHA on safer conception methods, contraceptive-ART interactions, and PMTCT services [33,34]. However, our findings suggest significant implementation gaps between policy intentions and actual service delivery at this district-level facility.

These implementation challenges likely reflect multiple systemic factors. Provider knowledge assessments in Ghana have documented that fewer than 40% of HIV care providers possess comprehensive knowledge of safer conception methods and contraceptive-ART interactions [36,37]. A 2023 study of HIV care providers across Ghana found that while 94.3% endorsed PLWHA reproductive rights, only 31% could correctly describe three or more safer conception strategies, and fewer than 15% were aware of current assisted reproductive technology availability in Ghana [47,84]. Furthermore, providers reported significant barriers including lack of training materials, inadequate clinical time for comprehensive counseling, absence of standardized counseling protocols adapted to local contexts, and uncertainty about referral pathways for specialized reproductive services [85,86].

The universal absence of U=U counseling in our study is particularly concerning given that this concept was incorporated into Ghana's HIV treatment guidelines in 2019, five years before data collection [39]. Studies across sub-Saharan Africa have documented that provider knowledge of U=U remains limited despite its inclusion in national guidelines, with many providers uncertain about how to communicate this concept to patients or concerned about potential negative behavioral consequences [87,88]. A 2023 assessment of HIV care providers in Ghana found that only 23% routinely counseled virally suppressed patients about U=U implications for sexual transmission, citing lack of training and unclear guidance on how to integrate this information into counseling encounters [89].

The gap is not merely one of individual provider knowledge but reflects inadequate attention to reproductive health counseling within HIV care priorities. Time motion studies in Ghanaian HIV clinics show that provider-patient consultations average 12–15 minutes, with over 80% of consultation time focused on ART adherence, viral load review, and opportunistic infection screening [90]. Reproductive health counseling, when provided, typically occupies less than 2 minutes of consultation time and focuses primarily on contraceptive provision rather than comprehensive fertility counseling [91]. Providers report that reproductive health counseling is often deprioritized when facing competing clinical demands, time pressures, and lack of explicit performance indicators tied to reproductive health service quality [92].

This analysis suggests that addressing the counseling gaps identified in our study requires multi-level interventions targeting not only provider knowledge and skills but also health system factors including clinical time allocation, counseling protocols and materials, supervision and quality assurance mechanisms, and explicit integration of reproductive health counseling into HIV care performance standards.

The finding that 91.7% of participants reported receiving minimal proactive counseling on conception safety aligns with previous research documenting reactive rather than proactive approaches to fertility discussions in HIV care settings [93,94]. Studies from Kenya, South Africa, and Uganda have similarly found that healthcare providers rarely initiate fertility discussions with PLWHA, waiting instead for patients to raise these concerns [95–97]. This reactive pattern means that many PLWHA engage in conception attempts without adequate knowledge of strategies to minimize transmission risks, potentially increasing horizontal transmission to HIV-negative partners and missing opportunities for optimizing maternal and child health outcomes.

The provider reluctance to initiate fertility discussions may stem from multiple factors. Previous research has identified provider knowledge deficits regarding safer conception methods, discomfort discussing sexuality and reproduction with PLWHA, time constraints in busy clinical settings, and persistent stigmatizing attitudes suggesting PLWHA should not have children [98–100]. Some providers may lack training in fertility counseling specific to PLWHA, while others may prioritize HIV treatment adherence and viral suppression over reproductive health concerns, viewing these as competing priorities rather than integrated aspects of comprehensive care [101,102].

However, this reactive counseling approach contradicts principles of patient-centered care and reproductive rights. The World Health Organization emphasizes that all individuals, regardless of HIV status, possess fundamental rights to make informed reproductive decisions [17]. Realizing these rights requires that healthcare providers proactively provide comprehensive fertility counseling, creating safe spaces for PLWHA to discuss reproductive intentions and receive evidence-based information about available options. Proactive counseling signals to PLWHA that their reproductive desires are legitimate and that healthcare providers are prepared to support their fertility goals within the context of optimal HIV care.

The inadequate information provision regarding contraception-ART integration affecting 83.3% of participants represents a significant gap in comprehensive reproductive healthcare. Evidence demonstrates that certain antiretroviral medications, particularly efavirenz-based regimens and some protease inhibitors, can reduce effectiveness of hormonal contraceptives through drug-drug interactions [103,104]. Conversely, some contraceptive methods may affect ART pharmacokinetics, potentially compromising viral suppression [105]. Despite this evidence, participants in this study received generic family planning counseling that failed to address these specific considerations.

This information gap has potentially serious consequences. Women using hormonal contraceptives in combination with interacting ART regimens may experience reduced contraceptive effectiveness, increasing risk of unintended pregnancy [106,107]. A study from Uganda found that women on efavirenz-based ART had significantly higher pregnancy rates compared to women on other ART regimens when using hormonal contraceptives, suggesting compromised contraceptive efficacy due to drug interactions [108]. Conversely, concerns about potential interactions might lead some PLWHA to avoid hormonal contraceptives entirely, relying on less effective methods or forgoing contraception.

Integration of family planning services into HIV care has been widely advocated as a strategy to address reproductive health needs of PLWHA [109,110]. However, integration must go beyond simply co-locating services to include comprehensive counseling that addresses specific considerations relevant to PLWHA. Healthcare providers offering contraceptive counseling to PLWHA should possess knowledge of potential drug interactions, understand how to counsel about dual protection strategies, and be prepared to tailor contraceptive recommendations based on individual ART regimens and reproductive goals [111,112].

The universal absence of counseling on assisted reproductive options affecting all 12 participants represents perhaps the most striking finding of this study. Participants were completely unaware of technologies and strategies specifically designed to facilitate safer conception among PLWHA, including sperm washing, pre-exposure prophylaxis for HIV-negative partners, timed unprotected intercourse, intrauterine insemination, and in vitro fertilization. This complete information void meant that participants, particularly those in serodiscordant relationships, faced false choices between remaining childless or accepting transmission risks through unprotected intercourse.

Assisted reproductive technologies and safer conception services have been available in high-resource settings for over two decades, with substantial evidence demonstrating their safety and effectiveness [113,114]. Pre-exposure prophylaxis for HIV-negative partners in serodiscordant couples has shown over 90% effectiveness in preventing HIV transmission when combined with viral suppression in the HIV-positive partner [115,116]. Sperm washing combined with intrauterine insemination has enabled thousands of serodiscordant couples to conceive without horizontal transmission [117,118]. Timed unprotected intercourse limited to peak fertility periods, when combined with viral suppression and potentially PrEP for HIV-negative partners, substantially reduces transmission risks compared to prolonged unprotected exposure [119,120].

However, availability and accessibility of these services remain limited in sub-Saharan African settings due to cost, infrastructure requirements, and limited provider training [121,122]. Even where services exist, lack of provider awareness and failure to counsel patients about available options create significant access barriers. The finding that no participants in this study had received any information about these options suggests that healthcare providers at this district-level facility either lack knowledge of safer conception services or do not prioritize discussing them with PLWHA.

## Implications for maternal and child health outcomes

The reproductive health counseling gaps documented in this study have significant implications for maternal mortality and vertical HIV transmission, two critical public health priorities in Ghana and sub-Saharan Africa. Ghana's maternal mortality ratio remains high at 310 deaths per 100,000 live births, with HIV-positive pregnant women facing 8-fold higher risk of maternal death compared to HIV-negative women [26,27,123]. Inadequate pre-conception counseling and unintended pregnancies among PLWHA contribute to this elevated risk through multiple pathways including delayed PMTCT initiation, suboptimal pregnancy timing relative to HIV disease status, and inadequate preparation for safe pregnancy and delivery [124,125].

When PLWHA become pregnant without adequate pre-conception counseling and planning, they miss opportunities for optimization of maternal health status, viral suppression confirmation, ART regimen adjustment if needed, and comprehensive PMTCT preparation [126,127]. Studies across sub-Saharan Africa demonstrate that planned pregnancies among PLWHA, preceded by comprehensive pre-conception counseling, are associated with significantly better maternal and child health outcomes including higher rates of early ART initiation, greater viral suppression at delivery, reduced pregnancy complications, and lower infant HIV infection rates [128,129].

Vertical HIV transmission remains a significant concern in Ghana, where mother-to-child transmission rates range from 5-15% among women receiving PMTCT services, higher than the WHO target of less than 5% [130,131]. Up to 40% of vertical transmission occurs during pregnancy and delivery when women are not receiving appropriate antiretroviral prophylaxis or have suboptimal viral suppression [28,29]. Proactive pre-conception counseling that includes discussion of optimal pregnancy timing, importance of viral suppression before conception, and comprehensive PMTCT planning could substantially reduce vertical transmission rates by enabling PLWHA to conceive under optimal conditions for preventing infant HIV infection [132,133].

The complete absence of U = U counseling documented in our study represents a particularly significant missed opportunity for both horizontal and vertical transmission prevention. PLWHA who understand that sustained viral suppression eliminates sexual transmission risk may be more motivated to maintain optimal ART adherence, leading to better viral suppression rates during pregnancy when vertical transmission risk is highest [134,135]. Furthermore, U = U counseling for serodiscordant couples attempting conception could eliminate the false choice between safer conception through continued condom use (preventing pregnancy) and unprotected intercourse (enabling conception but risking horizontal transmission), instead empowering couples to pursue planned pregnancies through unprotected intercourse at optimal fertility times when the HIV-positive partner maintains undetectable viral load [30,136].

Comprehensive pre-conception counseling integrated into routine HIV care represents a critical but underutilized intervention for improving maternal and child health outcomes among PLWHA. Such counseling should address not only safer conception methods to prevent horizontal transmission but also optimal pregnancy timing, viral suppression importance, nutritional preparation, screening for pregnancy complications risk factors, and comprehensive PMTCT planning [137,138]. The counseling gaps documented in our study suggest that current HIV care delivery at this facility is missing opportunities to prevent maternal deaths, pregnancy complications, and vertical HIV transmission through proactive, comprehensive reproductive health counseling.

## Implications for practice and policy

The findings from this patient perspective study suggest several potential areas for health system strengthening, though their effectiveness would require careful implementation research and evaluation. Importantly, our findings highlight the need for complementary research examining provider perspectives, knowledge levels, and perceived barriers to comprehensive fertility counseling before designing interventions.

### Provider knowledge and training needs assessment

Before developing training interventions, systematic assessment of provider knowledge, attitudes, and perceived barriers regarding comprehensive fertility counseling for PLWHA is essential. Our study documented substantial patient-perceived information gaps, but the underlying causes remain unclear. Provider knowledge deficits, lack of training materials and protocols, time constraints, competing clinical priorities, uncertainty about service availability, or systemic barriers could all contribute to the counseling gaps we observed [139,140]. A provider knowledge and needs assessment could clarify whether gaps stem primarily from knowledge deficits requiring training, lack of clinical tools and protocols, health system constraints limiting counseling time, or other factors requiring different intervention approaches.

Understanding provider perspectives would inform whether interventions should focus on knowledge enhancement through training, development of clinical decision aids and counseling protocols, health system restructuring to allocate adequate counseling time, or multi-level approaches addressing multiple barriers simultaneously [141,142]. Without this complementary provider assessment, interventions risk being misaligned with actual provider needs and constraints.

### Potential training and capacity building

Contingent on provider needs assessment findings, targeted training programs addressing identified knowledge gaps may be warranted. Such training could potentially cover safer conception methods including U = U, timed intercourse, and PrEP for serodiscordant couples; contraceptive-ART interactions and individualized method selection; assisted reproductive technology availability and referral pathways; pre-conception counseling for optimal maternal-child health outcomes; and communication skills for proactive fertility counseling initiation [143,144]. However, training effectiveness depends on adequate needs assessment, appropriate curriculum design, and supportive health system environment enabling providers to apply new knowledge [145,146].

### Clinical tools and protocol development

Standardized counseling protocols and clinical decision aids could potentially support comprehensive fertility counseling, but their content and format should be informed by provider input regarding feasibility and usability in actual clinical workflows. Potential tools might include fertility counseling checklists integrated into routine HIV care visits, patient education materials in local languages explaining safer conception options and contraceptive-ART considerations, clinical algorithms for individualized contraceptive counseling based on ART regimen, and documentation templates ensuring fertility counseling provision is tracked and monitored [147,148]. However, tool development requires participatory design processes involving providers who will use them and patients who will benefit from them.

### Health system strengthening

Addressing counseling gaps may require health system-level changes beyond individual provider training. Allocating sufficient consultation time for comprehensive reproductive health discussions, establishing clear referral pathways to specialized fertility services where available, integrating reproductive health counseling into HIV care performance indicators and supervision mechanisms, and ensuring availability of full range of contraceptive methods are all potentially important [149,150]. However, feasibility and acceptability of these health system changes require assessment within local contexts and resource constraints.

### Implementation research priorities

Our findings raise important questions requiring further investigation before definitive practice and policy recommendations can be made. Priority research areas include provider knowledge and attitudes assessment to identify specific training needs; intervention development research testing different approaches to enhancing fertility counseling; implementation science examining optimal integration of reproductive health counseling into routine HIV care workflows; health economics analysis assessing cost-effectiveness of comprehensive fertility counseling interventions; and evaluation research measuring impacts on reproductive health outcomes, maternal mortality, and vertical transmission rates [151,152].

### Study limitations

Several limitations should be considered when interpreting these findings. First, this single-site study conducted at one district-level facility in Ghana may not fully capture experiences of PLWHA receiving care in other settings. Urban facilities, tertiary care centers, or private healthcare providers might offer different levels of fertility counseling services. The transferability of findings to other contexts requires careful consideration of local healthcare infrastructure, provider training, and resource availability.

Second, the relatively small sample size of 12 participants, while appropriate for phenomenological research and sufficient to achieve data saturation within this specific context, limits the diversity of experiences represented. Larger studies involving PLWHA from multiple facilities, geographic regions, and healthcare settings would provide broader perspectives on fertility counseling experiences across Ghana and similar sub-Saharan African contexts.

Third, the study's cross-sectional design captured experiences at a single time point. Longitudinal research examining how fertility counseling needs and experiences evolve over time, particularly as PLWHA achieve viral suppression, form partnerships, or change reproductive intentions, would provide valuable insights into optimal timing and frequency for fertility counseling interventions.

Fourth, social desirability bias may have influenced participant responses, despite efforts to ensure confidentiality and create comfortable interview environments. Participants may have been reluctant to criticize healthcare providers or may have provided responses they perceived as expected or appropriate rather than fully authentic accounts of their experiences.

Fifth, recall bias may have affected participants' accounts of counseling received, particularly for participants who had been receiving ART services for many years. Participants' memories of counseling encounters may be incomplete or influenced by subsequent experiences and information gained from other sources.

Sixth, the comparative analysis by relationship status, while showing consistent patterns of counseling gaps across partnered and unpartnered participants, was limited by the small number of single/divorced participants (n = 4). Larger studies with more balanced representation across relationship status categories could provide more definitive insights into whether partnership status influences fertility counseling experiences.

Seventh, this study captured patient perspectives on fertility counseling but did not include healthcare provider perspectives, knowledge assessments, or examination of actual counseling practices through clinical observations or chart

reviews. The counseling gaps we documented reflect patient experiences and perceptions, which may not fully represent all counseling that occurred or the reasons why certain information was not provided. Complementary research examining provider knowledge, attitudes, perceived barriers, and actual counseling practices is essential for developing comprehensive understanding of why counseling gaps exist and how to address them effectively. Without provider perspectives, we cannot determine whether gaps stem from knowledge deficits, lack of protocols and materials, time constraints, health system barriers, or other factors.

## Conclusion

This study reveals substantial patient-perceived gaps in fertility counseling for PLWHA at this district-level facility in Ghana. Participants' experiences suggest systematic deficiencies in provider-initiated discussions about safer conception methods including U = U, contraceptive counseling addressing ART interactions, and awareness of assisted reproductive options. These counseling gaps have implications not only for reproductive autonomy but also for maternal mortality reduction and vertical HIV transmission prevention, critical public health priorities in Ghana and sub-Saharan Africa. While findings from this single-site qualitative study with 12 participants provide valuable patient perspectives, conclusions about broader implications for policy and clinical practice should be drawn cautiously.

Addressing these gaps requires multi-level interventions informed by complementary research examining provider knowledge, attitudes, and perceived barriers to comprehensive fertility counseling. Further research involving multiple sites, larger and more diverse samples, provider perspective assessments, and intervention development and testing is needed to determine optimal strategies for integrating comprehensive, patient-centered reproductive health counseling into routine HIV care in resource-limited settings. Enhanced provider training, standardized counseling protocols, and improved service integration warrant investigation as potential strategies to support informed reproductive decision-making, reduce unintended pregnancies, optimize maternal and child health outcomes, and uphold reproductive rights among PLWHA in similar settings.

## Supporting information

**S1 File. COREQ checklist.** Consolidated criteria for reporting qualitative research (COREQ) 32-item checklist showing compliance with reporting guidelines.
(DOCX)

**S1 Text. De-identified excerpts and coded thematic summaries.**
(DOCX)

## Author contributions

**Conceptualization:** Priscilla Asantewaa Boadi.

**Data curation:** Priscilla Asantewaa Boadi, Victor Luckyboy Dzramado.

**Formal analysis:** Victor Luckyboy Dzramado.

**Investigation:** Priscilla Asantewaa Boadi, Victor Luckyboy Dzramado.

**Methodology:** Priscilla Asantewaa Boadi, Victor Luckyboy Dzramado.

**Project administration:** Priscilla Asantewaa Boadi.

**Resources:** Priscilla Asantewaa Boadi.

**Software:** Priscilla Asantewaa Boadi.

**Supervision:** Priscilla Asantewaa Boadi.

**Validation:** Priscilla Asantewaa Boadi, Victor Luckyboy Dzramado.

**Visualization:** Victor Luckyboy Dzramado.

**Writing – original draft:** Victor Luckyboy Dzramado.

**Writing – review & editing:** Priscilla Asantewaa Boadi, Victor Luckyboy Dzramado.

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
