## [Decision Letter · Decision Letter 0]

12 Jan 2026

PGPH-D-25-04006

UNMET PATIENT NEEDS AND INFORMATION GAPS IN FERTILITY COUNSELING FOR PERSONS LIVING WITH HIV/AIDS: EVIDENCE FROM GHANA

Dear Dr. Dzramado,

Thank you for submitting your manuscript to PLOS Global Public Health. After careful consideration, we feel that it has merit but does not fully meet PLOS Global Public Health’s publication criteria as it currently stands. Therefore, we invite you to submit a revised version of the manuscript that addresses the points raised during the review process.

We look forward to receiving your revised manuscript.

Kind regards,

Md. Alamgir Hossain, M.S.S., M.P.H.

Academic Editor

Journal Requirements:

Please send a completed 'Competing Interests' statement, including any COIs declared by your co-authors. If you have no competing interests to declare, please state "The authors have declared that no competing interests exist".

Please amend your detailed Financial Disclosure statement. This is published with the article. It must therefore be completed in full sentences and contain the exact wording you wish to be published.If you did not receive any funding for this study, please simply state: “The authors received no specific funding for this work.”

In the online submission form, you indicated that “The datasets generated and analyzed during the current study are not publicly available to protect participant confidentiality but are available from the corresponding author on reasonable request.”All PLOS journals now require all data underlying the findings described in their manuscript to be freely available to other researchers, either1. In a public repository,2. Within the manuscript itself, or3. Uploaded as supplementary information.This policy applies to all data except where public deposition would breach compliance with the protocol approved by your research ethics board. If your data cannot be made publicly available for ethical or legal reasons (e.g., public availability would compromise patient privacy), please explain your reasons by return email and your exemption request will be escalated to the editor for approval. Your exemption request will be handled independently and will not hold up the peer review process, but will need to be resolved should your manuscript be accepted for publication. One of the Editorial team will then be in touch if there are any issues.

We have noticed that you have a list of Supporting Information legends in your manuscript. However, there are no corresponding files uploaded to the submission. Please upload them as separate files with the item type 'Supporting Information'.

Additional Editor Comments (if provided):

Reviewers' comments:

Reviewer's Responses to Questions

**Comments to the Author**

1. Does this manuscript meet PLOS Global Public Health’s publication criteria? Is the manuscript technically sound, and do the data support the conclusions? The manuscript must describe methodologically and ethically rigorous research with conclusions that are appropriately drawn based on the data presented.? Is the manuscript technically sound, and do the data support the conclusions? The manuscript must describe methodologically and ethically rigorous research with conclusions that are appropriately drawn based on the data presented.

Reviewer #1: Yes

Reviewer #2: Yes

2. Has the statistical analysis been performed appropriately and rigorously?

Reviewer #1: Yes

Reviewer #2: Yes

3. Have the authors made all data underlying the findings in their manuscript fully available (please refer to the Data Availability Statement at the start of the manuscript PDF file)?

The PLOS Data policy requires authors to make all data underlying the findings described in their manuscript fully available without restriction, with rare exception. The data should be provided as part of the manuscript or its supporting information, or deposited to a public repository. For example, in addition to summary statistics, the data points behind means, medians and variance measures should be available. If there are restrictions on publicly sharing data—e.g. participant privacy or use of data from a third party—those must be specified.requires authors to make all data underlying the findings described in their manuscript fully available without restriction, with rare exception. The data should be provided as part of the manuscript or its supporting information, or deposited to a public repository. For example, in addition to summary statistics, the data points behind means, medians and variance measures should be available. If there are restrictions on publicly sharing data—e.g. participant privacy or use of data from a third party—those must be specified.

Reviewer #1: No

Reviewer #2: No

4. Is the manuscript presented in an intelligible fashion and written in standard English?

Reviewer #1: Yes

Reviewer #2: Yes

Reviewer #1: This manuscript presents original and policy-relevant qualitative research examining unmet patient needs and information gaps in fertility counselling among persons living with HIV/AIDS in Ghana. The topic is highly relevant to global public health and reproductive health integration within HIV care, and the study design is broadly appropriate for addressing the stated objectives. The findings provide valuable patient-centered insights from an under-researched setting. However, several substantive issues must be addressed before the manuscript can be considered for publication. In particular, the authors should strengthen the following:

1. Methodological and Analytical Transparency

The qualitative design is appropriate, but further clarification is needed regarding:

• Rationale for sample size and saturation

• Coding procedures and reflexivity

2. Reporting Guideline Compliance

The manuscript does not explicitly state adherence to a qualitative reporting guideline (e.g., COREQ).

Explicit reporting guideline alignment is a PLOS requirement.

3. Ethics and Data Availability

• The inclusion of ethical approval in the abstract is appropriate; however, the statement could be strengthened by clearly indicating that informed consent was obtained from all participants, or by briefly clarifying the nature of the approval. It should also be more prominently and systematically presented.

• Data availability statement: While the statement provided is appropriate given the sensitive nature of the data, the statement does not fully meet PLOS requirements. The authors should better justify the restriction, specify what de-identified materials (e.g., transcripts or codebooks) can be shared, and clearly outline access conditions to demonstrate alignment with PLOS data availability and transparency standards. The authors should also identify the data custodian(s) and provide a clear contact address for data access requests.

4. Discussion and Interpretation

The discussion is generally strong but would benefit from:

• Deeper engagement with limitations (single-site study, small sample, transferability).

• More cautious framing of implications for policy and clinical practice.

Some conclusions verge on generalisation beyond the study context.

5. Language and Presentation

• The manuscript is written in good academic English but would benefit from tightening, particularly in the abstract and discussion.

Reviewer #2: This manuscript is well written and highly informative, and I thoroughly enjoyed reading it. The authors clearly identified an important gap in the clinical care of people living with HIV/AIDS (PLWHA) in a district-level facility in Ghana and designed a study that appropriately investigates this issue. The findings are strong and have the potential to inform local health professionals and policymakers in improving health professions education and training, as well as service delivery through a more holistic and patient-centered approach. Importantly, the results are not only applicable to similar settings within Ghana but are also transferable to district-level facilities across many sub-Saharan African countries.

I have one minor comment. In the Methods section, the authors state that they collected field notes documenting nonverbal cues, emotional reactions, and contextual observations; however, it is unclear how data from these methods were analyzed or whether they contributed to the results. Readers would benefit from greater clarity on how these data were analyzed and what insights they generated, particularly to support replication in similar contexts. Alternatively, if these data did not meaningfully contribute to the findings, the authors may consider clarifying this or removing this component to avoid confusion.

**Do you want your identity to be public for this peer review?** For information about this choice, including consent withdrawal, please see our Privacy Policy..

Reviewer #1: No

Reviewer #2: No

---

## [Decision Letter · Decision Letter 1]

15 Mar 2026

PGPH-D-25-04006R1

UNMET PATIENT NEEDS AND INFORMATION GAPS IN FERTILITY COUNSELING FOR PERSONS LIVING WITH HIV/AIDS: EVIDENCE FROM GHANA

Dear Dr. Victor Luckyboy Dzramado,

Thank you for submitting your manuscript to PLOS Global Public Health. After careful consideration, we feel that it has merit but does not fully meet PLOS Global Public Health’s publication criteria as it currently stands. Therefore, we invite you to submit a revised version of the manuscript that addresses the points raised during the review process.

We look forward to receiving your revised manuscript.

Kind regards,

Md. Alamgir Hossain, M.S.S., M.P.H.

Academic Editor

Journal Requirements:

Additional Editor Comments (if provided):

Reviewers' comments:

Reviewer's Responses to Questions

**Comments to the Author**

Reviewer #3: (No Response)

Reviewer #4: All comments have been addressed

publication criteria? Is the manuscript technically sound, and do the data support the conclusions? The manuscript must describe methodologically and ethically rigorous research with conclusions that are appropriately drawn based on the data presented.? Is the manuscript technically sound, and do the data support the conclusions? The manuscript must describe methodologically and ethically rigorous research with conclusions that are appropriately drawn based on the data presented.

Reviewer #3: (No Response)

Reviewer #4: Yes

3. Has the statistical analysis been performed appropriately and rigorously?

Reviewer #3: (No Response)

Reviewer #4: Yes

4. Have the authors made all data underlying the findings in their manuscript fully available (please refer to the Data Availability Statement at the start of the manuscript PDF file)?

The PLOS Data policy requires authors to make all data underlying the findings described in their manuscript fully available without restriction, with rare exception. The data should be provided as part of the manuscript or its supporting information, or deposited to a public repository. For example, in addition to summary statistics, the data points behind means, medians and variance measures should be available. If there are restrictions on publicly sharing data—e.g. participant privacy or use of data from a third party—those must be specified.requires authors to make all data underlying the findings described in their manuscript fully available without restriction, with rare exception. The data should be provided as part of the manuscript or its supporting information, or deposited to a public repository. For example, in addition to summary statistics, the data points behind means, medians and variance measures should be available. If there are restrictions on publicly sharing data—e.g. participant privacy or use of data from a third party—those must be specified.

Reviewer #3: (No Response)

Reviewer #4: No

5. Is the manuscript presented in an intelligible fashion and written in standard English?

Reviewer #3: (No Response)

Reviewer #4: Yes

Reviewer #3: 1. Please clarify how PLWHA is defined in the context of this study (e.g., diagnosed individuals, those on ART, adults only).

2. The paragraph referencing reproductive rights cites WHO but does not explicitly link to: Ghana national reproductive policy, National HIV strategic framework. Explain national fertility rate and PLWHA fertility rate.

3. The study claims a phenomenological approach but does not specify the philosophical alignment (e.g., Husserlian, Heideggerian, Colaizzi etc). OR author may delete the word “ phenomenological”.

4. Please specify whether interviews were conducted face-to-face and describe in the interview setting. Clarify whether any participant incentives or reimbursements were provided.

5. Although 450 ART clients were active during data collection, only 12 were included. Please explain: How participants were selected, How many were approached, How many had refused, How saturation was determined.

6. Provide clearer explanation of why purposive sampling was used and how it differed from convenience sampling in practice. Also mention in limitation section as both have their own bias.

7. The inclusion of confidence intervals appears inconsistent with phenomenological methodology. Please justify or reconsider their use.

8. Clarify who runs the ART clinic and whether safer conception counseling is part of routine services. Since hospital was well empowered by manpower why counseling part was lagging. Reason would strengthen your findings.

9. The discussion mentions evidence that efavirenz-based ART reduces contraceptive effectiveness. Was ART regimen data collected for the 12 participants in the study?

Reviewer #4: My review has been submitted as an attachment

**Do you want your identity to be public for this peer review?** For information about this choice, including consent withdrawal, please see our Privacy Policy..

Reviewer #3: **Yes:** DR.NIRAZ YADAVDR.NIRAZ YADAVDR.NIRAZ YADAVDR.NIRAZ YADAV

Reviewer #4: No

---

## [Editor Report · Decision Letter 2]

18 Mar 2026

PGPH-D-25-04006R2

UNMET PATIENT NEEDS AND INFORMATION GAPS IN FERTILITY COUNSELING FOR PERSONS LIVING WITH HIV/AIDS: EVIDENCE FROM GHANA

Dear Dr. Victor Luckyboy Dzramado,

Thank you for submitting your manuscript to PLOS Global Public Health. After careful consideration, we feel that it has merit but does not fully meet PLOS Global Public Health’s publication criteria as it currently stands. Therefore, we invite you to submit a revised version of the manuscript that addresses the points raised during the review process.

We look forward to receiving your revised manuscript.

Kind regards,

Md. Alamgir Hossain, M.S.S., M.P.H.

Academic Editor
---

## [Editor Report · Decision Letter 3]

24 Mar 2026

UNMET PATIENT NEEDS AND INFORMATION GAPS IN FERTILITY COUNSELING FOR PERSONS LIVING WITH HIV/AIDS: EVIDENCE FROM GHANA

PGPH-D-25-04006R3

Dear Dzramado,

We are pleased to inform you that your manuscript 'UNMET PATIENT NEEDS AND INFORMATION GAPS IN FERTILITY COUNSELING FOR PERSONS LIVING WITH HIV/AIDS: EVIDENCE FROM GHANA' has been provisionally accepted for publication in PLOS Global Public Health.

Best regards,

Md. Alamgir Hossain, M.S.S., M.P.H.

Academic Editor